# DEEP ENSEMBLES: A LOSS LANDSCAPE PERSPECTIVE

## ABSTRACT

Deep ensembles have been empirically shown to be a promising approach for improving accuracy, uncertainty and out-of-distribution robustness of deep learning models. While deep ensembles were theoretically motivated by the bootstrap, non-bootstrap ensembles trained with just random initialization also perform well in practice, which suggests that there could be other explanations for why deep ensembles work well. Bayesian neural networks, which learn distributions over the parameters of the network, are theoretically well-motivated by Bayesian principles, but do not perform as well as deep ensembles in practice, particularly under dataset shift. One possible explanation for this gap between theory and practice is that popular scalable approximate Bayesian methods tend to focus on a single mode, whereas deep ensembles tend to explore diverse modes in function space. We investigate this hypothesis by building on recent work on understanding the loss landscape of neural networks and adding our own exploration to measure the similarity of functions in the space of predictions. Our results show that random initializations explore entirely different modes, while functions along an optimization trajectory or sampled from the subspace thereof cluster within a single mode predictions-wise, while often deviating significantly in the weight space. We demonstrate that while low-loss connectors between modes exist, they are not connected in the space of predictions. Developing the concept of the diversity–accuracy plane, we show that the decorrelation power of random initializations is unmatched by popular subspace sampling methods.

## 1 INTRODUCTION

Consider a typical classification problem, where $\boldsymbol{x}_n \in \mathbb{R}^D$ denotes the $D$-dimensional features and $y_n \in [1, \ldots, K]$ denotes the class label. Assume we have a parametric model $p(y|\boldsymbol{x}, \boldsymbol{\theta})$ for the conditional distribution where $\boldsymbol{\theta}$ denotes weights and biases of a neural network, and $p(\boldsymbol{\theta})$ is a prior distribution over parameters. The Bayesian posterior over parameters is given by

$$p(\boldsymbol{\theta}|\{\boldsymbol{x}_n, y_n\}_{n=1}^N) \propto p(\boldsymbol{\theta}) \prod_{n=1}^N p(y_n|\boldsymbol{x}_n, \boldsymbol{\theta}). \tag{1}$$

Computing the exact posterior distribution over $\boldsymbol{\theta}$ is computationally expensive (if not impossible) when $p(y_n|\boldsymbol{x}_n, \boldsymbol{\theta})$ is a deep neural network. A variety of approximations have been developed for *Bayesian neural networks*, including Laplace approximation (MacKay, 1992), Markov chain Monte Carlo methods (Neal, 1996; Welling & Teh, 2011; Springenberg et al., 2016), variational Bayesian methods (Graves, 2011; Blundell et al., 2015; Louizos & Welling, 2017; Wen et al., 2018) and Monte-Carlo dropout (Gal & Ghahramani, 2016; Srivastava et al., 2014). While computing the posterior is challenging, it is usually easy to perform maximum-a-posteriori (MAP) estimation, which corresponds to a mode of the posterior. The MAP solution can be written as the minimizer of the following loss (negative log likelihood + negative log prior):

$$\hat{\boldsymbol{\theta}}_{\mathsf{MAP}} = \arg\min_{\boldsymbol{\theta}} L(\boldsymbol{\theta}, \{\boldsymbol{x}_n, y_n\}_{n=1}^N) = \arg\min_{\boldsymbol{\theta}} -\log p(\boldsymbol{\theta}) - \sum_{n=1}^N \log p(y_n|\boldsymbol{x}_n, \boldsymbol{\theta}). \tag{2}$$

The MAP solution is computationally efficient, but only gives a point estimate and not a distribution over parameters. *Deep ensembles*, proposed by Lakshminarayanan et al. (2017), train an ensemble

of neural networks by initializing at $M$ different values and repeating the minimization multiple times which could lead to $M$ different solutions, if the loss is non-convex. (Lakshminarayanan et al. (2017) found adversarial training provides additional benefits in some of their experiments, but we will ignore adversarial training and focus only on ensembles with random initialization in this paper.)

Given finite training data, many parameter values could equally well explain the observations, and capturing these diverse solutions is crucial for quantifying *epistemic uncertainty* (Kendall & Gal, 2017). Bayesian neural networks learn a distribution over weights, and a good posterior approximation should be able to learn multi-modal posterior distributions in theory. Deep ensembles were inspired by the bootstrap (Breiman, 1996), which has nice theoretical properties. However, it has been empirically observed by Lakshminarayanan et al. (2017); Lee et al. (2015) that training individual networks with just random initialization is sufficient in practice and using the bootstrap even hurts performance in some cases (e.g. for small ensemble sizes). Furthermore, Ovadia et al. (2019) and Gustafsson et al. (2019) independently benchmarked existing methods for uncertainty quantification on a variety of datasets and architectures, and observed that ensembles tend to outperform approximate Bayesian neural networks in terms of both accuracy and uncertainty, particularly under dataset shift.

These empirical observations raise an important question: *Why do ensembles trained with just random initialization work so well in practice?* One possible hypothesis is that ensembles tend to sample from different modes[1] in function space, whereas variational Bayesian methods (which minimize $D_{\mathrm{KL}}(q(\boldsymbol{\theta})|p(\boldsymbol{\theta}|\{\boldsymbol{x}_n, y_n\}_{n=1}^N))$) might fail to explore multiple modes even though they are effective at capturing uncertainty within a single mode. See Figure 1 for a cartoon illustration. Note that while the MAP solution is

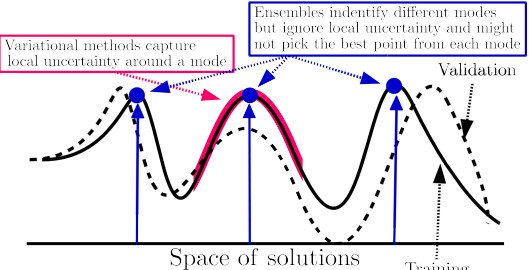

Figure 1: Cartoon illustration of the hypothesis. $x$-axis indicates parameter values and $y$-axis plots the negative loss $-L(\boldsymbol{\theta}, \{\boldsymbol{x}_n, y_n\}_{n=1}^N)$ on train and validation data.

a local minima for the training loss by definition, it may not necessarily be a local minima for the validation loss.

Recent work on understanding loss landscapes (Fort & Jastrzebski, 2019; Draxler et al., 2018; Garipov et al., 2018) allows us to investigate this hypothesis. Note that prior work on loss landscapes has focused on mode-connectivity and low-loss tunnels, but has not explicitly focused on how diverse the functions from different modes are, beyond an initial exploration in Fort & Jastrzebski (2019). Our findings show that:

- The functions sampled along a single training trajectory or subspace thereof (e.g. diagonal Gaussian, low-rank Gaussian and Dropout subspaces) tend to be very similar in predictions (while potential far away in the weight space), whereas functions sampled from different randomly initialized trajectories tend to be very diverse.

- Solution modes are connected in the loss landscape but they are distinct in the space of predictions. Low-loss tunnels create functions with near-identical low values of loss along the path, however these functions tend to be very different in function space, changing significantly in the middle of the tunnel.

## 2 BACKGROUND

The loss landscape of neural networks (also called the objective landscape) – the space of weights and biases that the network navigates – is typically a very high dimensional function and therefore could potentially be very complicated. However, many empirical results show interesting properties of the loss surface. Goodfellow & Vinyals (2014) observed that the loss along a linear path from an initialization to the corresponding optimum is monotonically decreasing, encountering no significant

---

[1]We use the term mode to refer to unique functions $f_{\boldsymbol{\theta}}(\boldsymbol{x})$. Due to weight space symmetries, different parameters could correspond to the same function, i.e. $f_{\boldsymbol{\theta}_1}(\boldsymbol{x}) = f_{\boldsymbol{\theta}_2}(\boldsymbol{x})$ even though $\boldsymbol{\theta}_1 \neq \boldsymbol{\theta}_2$, but we ignore this aspect and leave it to future work.

obstacles along the way. Li et al. (2018) demonstrated that constraining optimization to a random, low-dimensional hyperplane in the weight space leads to results comparable to full-space optimization, provided that the dimension exceeds a modest threshold. This was geometrically understood and extended in (Fort & Scherlis, 2019). Garipov et al. (2018); Draxler et al. (2018) demonstrate that while a linear path between two independent optima hits a high loss area in the middle, there in fact exist continuous, low-loss paths connecting any pair of optima. These observations are unified into a single phenomenological model in (Fort & Jastrzebski, 2019). While independent, low-loss optima in the loss landscape are connected, Fort & Jastrzebski (2019) provide an early indication that in fact they represent very different functions in terms of their predictions. Therefore the connectivity cannot be due to trivial symmetries of the network which would keep the input–output mapping intact.

## 3 Visualizing Function Similarity Across Initializations

We train convolutional neural networks on the CIFAR-10 (Krizhevsky, 2009) dataset:

- *SmallCNN*: channels [16,32,32] for 10 epochs which achieves $64\%$ test accuracy.
- *MediumCNN*: channels [64,128,256,256] for 20 epochs which achieves $70\%$ test accuracy.
- *ResNet20v1*: for 200 epochs which achieves $90\%$ test accuracy.

We use the Adam optimizer (Kingma & Ba, 2015) for training and to make sure the effects we observe are general, we validate that our results hold for vanilla stochastic gradient descent (SGD) as well, which we do not show in this paper. We use batch size 128 and dropout 0.03 for training *SmallCNN* and *MediumCNN*. To generate weight space and prediction space similarity results, we use a constant learning rate of $1.6 \times 10^{-3}$, unless specified otherwise. We do not use any data augmentation with those two architectures. For *ResNet20v1*, we use the data augmentation and learning rate schedule used in Keras examples[2]. The overall trends are consistent across all architectures, datasets, and other hyperparameter and non-linearity choices we explored.

### 3.1 Similarity of Functions Within and Across Trajectories

First, we compute the similarity between different checkpoints along a single trajectory. We plot the cosine similarity in weight space in Figure 2(a) and the disagreement in function space, defined as the fraction of points the checkpoints disagree on, in Figure 2(b). We observe that the checkpoints along a trajectory are largely similar both in the weight space and the function space. Next, we evaluate how diverse the final solutions from different random initializations are. The functions from different initialization are different, as demonstrated by the similarity plots in Figure 3. Comparing this with Figures 2(a) and 2(b), we see that functions within a single trajectory exhibit higher similarity and functions across different trajectories exhibit much lower similarity.

Next, we take the predictions from different checkpoints along the individual training trajectories from multiple initializations and compute a t-SNE plot (Maaten & Hinton, 2008) to visualize their similarity in function space. More precisely, we take the softmax output for a set of points, flatten the vector and use it as the input to the t-SNE plot. Figure 2(c) shows that the functions explored by different trajectories (denoted by circles with different colors) are far away, while functions explored within a single trajectory (circles with the same color) tend to be much more similar.

### 3.2 Similarity of Functions Across Subspaces from Each Trajectory

In addition to the checkpoints along a trajectory, we also construct subspaces based on each individual trajectory. Scalable Bayesian methods typically compute statistics based on the weights along a trajectory, hence visualizing the diversity of functions between the subspace helps understand the difference between Bayesian neural networks and ensembles. We use a representative set of four subspace sampling methods: a random subspace, a Monte Carlo dropout, a diagonal Gaussian approximation, and a low-rank covariance matrix Gaussian approximation. In the descriptions of the methods, let $\vec{w}_0$ be the current weight-space position (the weights and biases of our trained neural net) around which we will construct the subspace.

---

[2] https://keras.io/examples/cifar10_resnet/

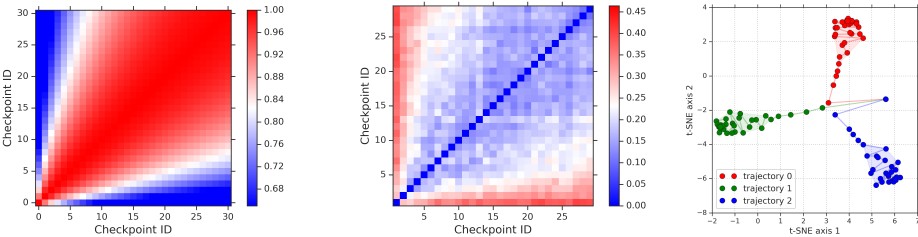

(a) Cosine similarity (weight space)  (b) Disagreement (prediction space)  (c) t-SNE of predictions

Figure 2: Results using SimpleCNN on CIFAR-10. *Left plot*: Cosine similarity between checkpoints to measure weight space alignment along optimization trajectory. *Middle plot*: The fraction of labels on which the predictions from different checkpoints disagree. *Right plot*: t-SNE plot of predictions from checkpoints corresponding to 3 different randomly initialized trajectories (in different colors).

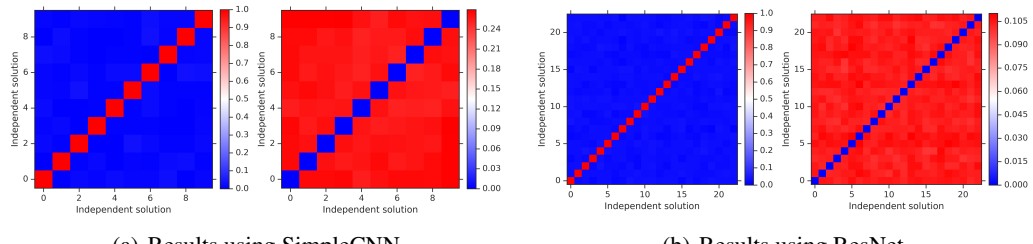

(a) Results using SimpleCNN                    (b) Results using ResNet

Figure 3: Results on CIFAR-10 using two different architectures. For each of these architectures, the left subplot shows the cosine similarity between different solutions in weight space, and the right subplot shows the fraction of labels on which the predictions from different solutions disagree.

- **Random subspace sampling**: We start at an optimized solution $\vec{w}_0$ and choose a random direction $\hat{v}$ in the weight space. We step in that direction by choosing different values of $t$ and looking at predictions at configurations $\vec{w}_0 + t\hat{v}$. We do this for many random directions $\hat{v}$.

- **Monte Carlo dropout subspace**: We start at an optimized solution $\vec{w}_0$ and apply dropout with a randomly chosen $p_{\text{keep}}$ to it. We do this many times, each time choosing a random $p_{\text{keep}}$, and look at predictions at $\text{dropout}_{p_{\text{keep}}}(\vec{w}_0)$.

- **Diagonal Gaussian subspace**: We start at an optimized solution $\vec{w}_0$ and look at the most recent iterations of training proceeding it. For each trainable parameter $w_i$, we calculate its mean $\text{mean}_i$ and standard deviation $\text{std}(w_i)$. To sample solutions from the subspace, we draw each parameter independently as $w_i \sim \mathcal{N}(\text{mean}_i, \text{std}_i)$. We repeat this many times and obtain predictions in each. This corresponds to sampling from a normal distribution with a diagonal covariance matrix.

- **Low-rank Gaussian subspace**: We start at an optimized solution $\vec{w}_0$ and look at the most recent iterations of training proceeding it. For each trainable parameter $w_i$, we calculate its mean $\text{mean}_i$. For a rank-$k$ approximation, we calculate top $k$ principal components of the weight vectors in the most recent iterations of training $\{\vec{p}_i \in \mathbb{R}^{\text{params}}\}_k$. We sample from a $k$-dimensional normal distribution and obtain the weight configurations as $\vec{w} \sim \vec{\text{mean}} + \sum_i \mathcal{N}(0^k, 1^k)\vec{p}_i$.

Figure 4 shows that functions sampled from a subspace (denoted by colored squares) corresponding to a particular initialization, are much more similar to each other. While some subspaces are more diverse, they still do not overlap with functions from another randomly initialized trajectory.

**Diversity versus Accuracy plots** To illustrate the difference in another fashion, we sample functions from a single subspace and plot diversity (as measured by disagreement between predictions) versus accuracy in Figure 5. Comparing these subspace points (colored dots) to the baseline optima (green star) and the optima from different random initializations (denoted by red stars), we observe that random initializations are much more effective at sampling diverse and accurate solutions, than subspace based methods constructed out of a single trajectory.

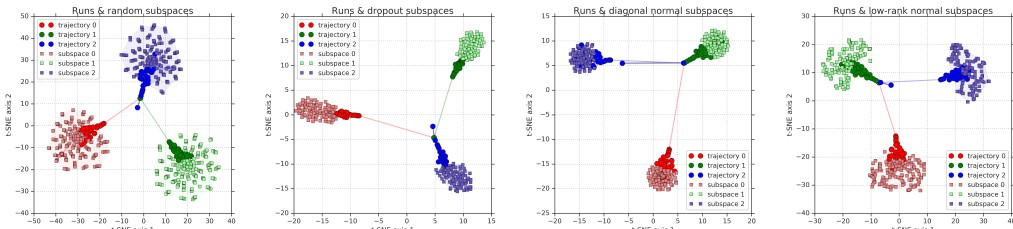

Figure 4: *Results using SimpleCNN on CIFAR-10*: t-SNE plots of validation set predictions for each trajectory along with four different subspace generation methods (showed by squares), in addition to 3 independently initialized and trained runs (different colors). As visible in the plot, the subspace-sampled functions stay in the prediction-space neighborhood of the run around which they were constructed, demonstrating that truly different functions are not sampled.

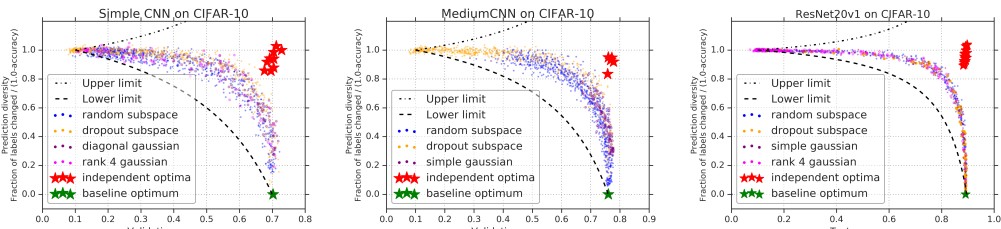

Figure 5: *Diversity versus accuracy plots for 3 models trained on CIFAR-10: SmallCNN, Medium-CNN and a ResNet20v1.* The clear separation between the subspace sampling populations (for 4 different subspace sampling methods) and the population of independently initialized and optimized solutions (red stars) is visible. The 2 limiting curves correspond to solution generated by perturbing the reference solution's predictions (bottom curve) and completely random predictions at a given accuracy (upper curve).

The diversity score used above quantifies the difference of two functions, by measuring fraction of points on which their predictions differ. We chose this approach due to its simplicity; one could also compute the KL-divergence or other distances between the output probability distributions Let $d_{\mathrm{diff}}$ denote the fraction of predictions on which the two functions differ. It is $0$ when the two functions make identical class predictions, and $1$ when they differ on every single example. To account for the fact that the lower the accuracy of a function, the higher its potential $d_{\mathrm{diff}}$ due to the possibility of the wrong answers being random and uncorrelated between the two functions, we normalize this by $(1-a)$, where $a$ is the accuracy.

For a reference function $f^*$ of accuracy $a^*$ and a function $f$ of accuracy $a$ whose predictions are obtained by *randomly perturbing* the predictions of $f^*$, the expected fractional difference is $d_{\mathrm{diff}} = (C-1)(a^*-a)/(a^*C-1)$, where $C$ is the number of classes. If the function $f$ of accuracy $a$ were entirely independent of $f^*$, then the expected fractional difference would be $d_{\mathrm{diff}} = (1-a^*)a + (1-a)a^* + (1-a^*)(1-a)(C-2)/(C-1)$. Those two limiting behaviours – the function $f$ being derived from $f^*$ by a perturbation, and $f$ and $f^*$ being completely independent – form the two dashed lines in Figure 5. We refer to Appendix D for further details on the limiting curves. The diversity reached is not as high as the theoretical optimum even for the independently initialized and optimized solutions, which provides scope for future work.

### 3.3 IDENTICAL LOSS DOES NOT IMPLY IDENTICAL FUNCTIONS IN PREDICTION SPACE

Figure 6 shows the radial loss landscape (train as well as the validation set) along the directions of two different optima. The left subplot shows that different trajectories achieve similar values of the loss, and the right subplot shows the similarity of these functions to their respective optima (in particular the fraction of labels predicted on which they differ divided by their error rate). While the loss values from different optima are similar, the functions are different, which confirms that random initialization leads to different modes in function space.

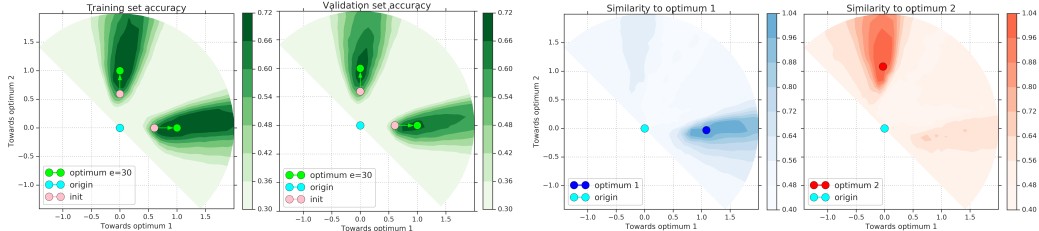

Figure 6: *Results using MediumCNN on CIFAR-10*: Radial loss landscape cut between the origin and two independent optima and the predictions of models on the same plane.

We construct a low-loss tunnel between different optima using the procedure proposed by Fort & Jastrzebski (2019), which is a simplification of the procedures proposed in Garipov et al. (2018) and Draxler et al. (2018). As shown in Figure 7(a), we start at the linear interpolation point (denoted by the black line) and reach the closest point on the manifold by minimizing the training loss. The minima of the training loss are denoted by the yellow line in the manifolds. Figure 7(b) confirms that the tunnel is indeed low-loss.

In order to visualize the 2-dimensional cut through the loss landscape and the the associated predictions on along a curved low-loss path, we divide the path into linear segments, and compute the loss and prediction similarities on a triangle given by this segment on one side and the origin of the weight space on the other. We perform this operation on each of the linear segments from which the low-loss path is constructed, and place them next to each other for visualization. Figure 8 visualizes the loss along the manifold, as well as the similarity to the original optima. Note that the regions between radial yellow lines consist of segments, and we stitch these segments together in Figure 8. The accuracy plots show that as we traverse along the low-loss tunnel, the accuracy remains fairly constant as expected. However, the prediction similarity plot shows that the low-loss tunnel does not correspond to similar solutions in function space. What it shows is that while the modes are connected in terms of accuracy/loss, their functional forms remain distinct and they do not collapse into a single mode.

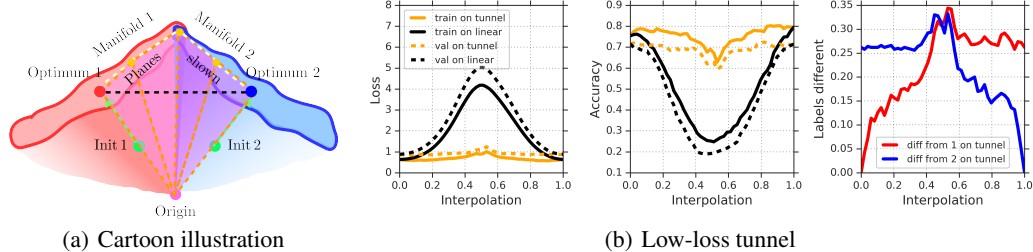

(a) Cartoon illustration          (b) Low-loss tunnel

Figure 7: *Left*: Cartoon illustration showing linear connector (black) along with the optimized connector which lies on the manifold of low loss solutions. *Right*: The loss and accuracy in between two independent optima on a linear path and an optimized path in the weight space.

## 4   EVALUATING THE RELATIVE EFFECTS OF ENSEMBLING VERSUS SUBSPACE METHODS

Our observations in the previous section suggest that subspace-based methods and ensembling should provide complementary benefits in terms of uncertainty and accuracy. To test this, we evaluate the performance of the following four variants using SmallCNN on CIFAR-10:

- *Baseline*: optimum at the end of a single training trajectory.
- *Subspace sampling*: average predictions over the solutions sampled from a subspace.
- *Ensemble*: train baseline multiple times with random initialization and average the predictions.
- *Ensemble + Subspace sampling*: train multiple times with random initialization, use subspace sampling within each trajectory.

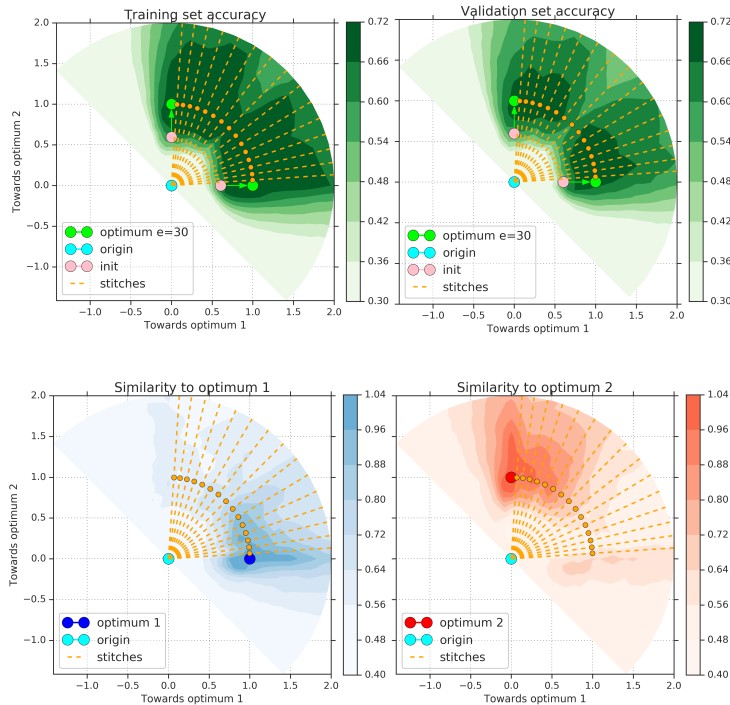

Figure 8: *Results using MediumCNN on CIFAR-10*: Radial loss landscape cut between the origin and two independent optima along an optimize low-loss connector and predictions similarity along the same planes.

Figures 9(a) and 9(b) show the results for low rank Gaussian subspace and diagonal Gaussian subspace respectively. The results validate our hypothesis as (i) subspace sampling and ensembling provide complementary benefits, and (ii) the relative benefits of ensembling are higher as it averages predictions over more diverse solutions.

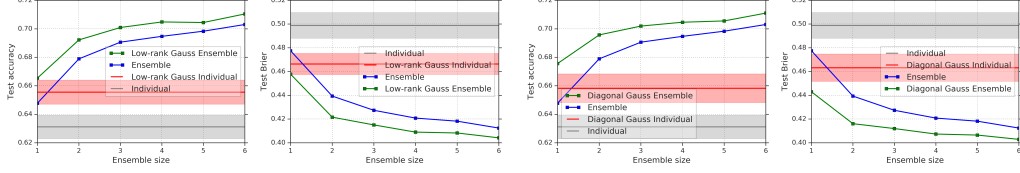

(a) Low rank Gaussian: Accuracy and Brier  (b) Diagonal Gaussian: Accuracy and Brier

Figure 9: Results on CIFAR-10 showing the complementary benefits of ensemble and subspace methods, as well as the effect of ensemble size.

**Weight averaging within a subspace** One could use the mean and diagonal/low-rank variance to approximate each mode of the posterior, however that increases the number of parameters required for each mode. Using just the mean weight for each mode would not increase the number of parameters. Izmailov et al. (2018) proposed stochastic weight averaging (SWA) for better generalization. One could also compute an (exponential moving) average of the weights along the trajectory, inspired by Polyak-Ruppert averaging in convex optimization, (see also (Mandt et al., 2017) for a Bayesian view on iterate averaging). As weight averaging has been already studied by Izmailov et al. (2018), we do not discuss it in detail. Figure S1 provides an illustration of why these strategies might help with generalization. We use weight averaging (WA) on the last few epochs which corresponds to using the mean of the subspace within each mode. Figure 10(a) shows that weight averaging achieves better performance within each mode, and ensemble + WA performs as well as ensemble + subspace combination methods, without any additional parameter overhead.

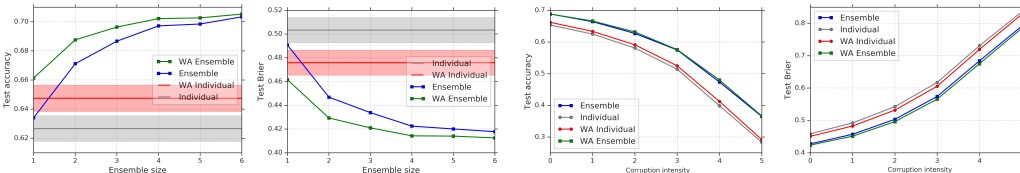

(a) Accuracy & Brier: Weight Averaging vs. Ensemble (b) Results on CIFAR-10-C: Accuracy & Brier versus Corruption Intensity

Figure 10: Results on CIFAR-10 using SimpleCNN: clean test and CIFAR-10-C corrupted test set.

Figure 10(b) shows accuracy and Brier score on CIFAR-10, both on the usual test set (corresponding to the intensity = 0 column) as well as on the CIFAR-10-C benchmark proposed (Hendrycks & Dieterich, 2019) which contains corrupted versions of CIFAR-10 with varying intensity values (1-5), making it useful to verify calibration under dataset shift (Ovadia et al., 2019). We see that ensembling and weight-averaging provide complementary benefits. WA improves over the vanilla baseline, but combining WA with ensembling over multiple random initializations improves performance further. Figure 9 reports accuracy and Brier score on the usual CIFAR-10 test set as a function of ensemble size. Under dataset shift, it is particular important to have diverse functions to avoid overconfident predictions (as averaging over similar functions would not reduce overconfidence).

## 4.1 RESULTS ON IMAGENET

To illustrate the effect on another challenging dataset, we repeat these experiments on ImageNet (Deng et al., 2009) using the same ResNet20V1 architecture. Due to computational constraints, we focus mainly on the experiment decoupling the effect of weight averaging vs ensembling. Figure 11(a) shows the complementary effects of ensembling and weight averaging; Figure 11(b) shows results on (subset of) ImageNet-C demonstrating that these trends are similar to those observed on CIFAR-10.

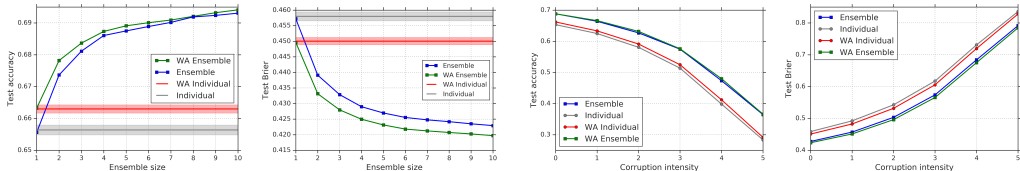

(a) Accuracy & Brier: Weight Averaging vs. Ensemble (b) Results on ImageNet-C: Accuracy & Brier versus Corruption Intensity

Figure 11: Results using ResNet on ImageNet: clean test and ImageNet-C corrupted test set.

## 5 DISCUSSION

Our results show that trajectories of randomly initialized neural networks explore different modes in function space, which explains why deep ensembles with random initializations help. They are essentially orthogonal to each other in the space of weights and very diverse in terms of their predictions. While these modes can be connected via optimized low-loss paths between them, we demonstrate that they correspond to distinct functions in terms of their predictions. Therefore the connectivity in the loss landscape does not imply connectivity in the space of functions.

Subspace sampling methods such as weight averaging, Monte Carlo dropout, and various versions of local Gaussian approximations, sample functions that might lie relatively far from the starting point in the weight space, however, they remain in the vicinity of their starting point in terms of predictions, giving rise to an insufficiently diverse set of functions. Using the concept of the diversity–accuracy plane, we demonstrate empirically that these subspace sampling methods never reach the combination of diversity and accuracy that independently trained models do, limiting their usefulness for ensembling.

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

# A   VISUALIZING THE LOSS LANDSCAPE ALONG ORIGINAL DIRECTIONS AND WA DIRECTIONS

Figure S1 shows the loss landscape (train as well as the validation set) and the effect of WA.

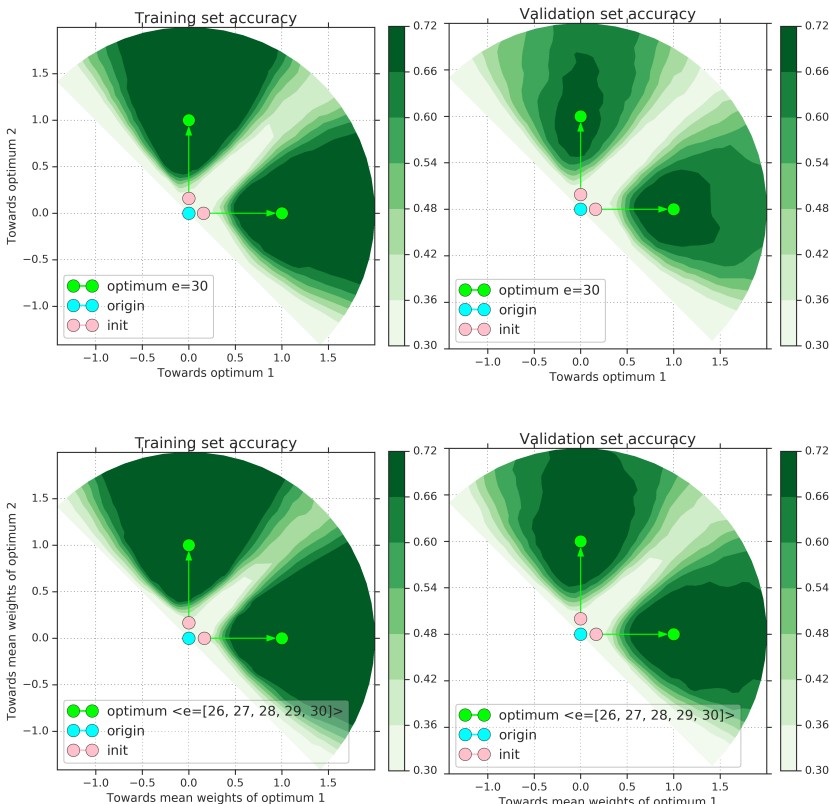

Figure S1: *Loss landscape versus generalization*: weights are typically initialized close to 0 and increase radially through the course of training. *Top row*: we pick two optima from different trajectories as the axes, and plot loss surface. Looking at $x$ and $y$ axes, we observe that while a wide range of radii achieve low loss on training set, the range of optimal radius values is narrower on validation set. *Bottom row*: we average weights within each trajectory using WA and use them as axes. A wider range of radius values generalize better along the WA directions, which confirms the findings of Izmailov et al. (2018).

# B   ADDITIONAL ABLATION EXPERIMENTS

## B.1   EFFECT OF RANDOMNESS: RANDOM INITIALIZATION VERSUS RANDOM SHUFFLING

Random seed affects both initial parameter values as well the order of shuffling of data points. We run experiments to decouple the effect of random initialization and shuffling; Figure S2 shows shows the results. We observe that both of them provide complementary sources of randomness, with random initialization being the dominant of the two. As expected, random mini-batch shuffling adds more randomness at higher learning rates due to gradient noise.

# C   ADDITIONAL DIVERSITY − ACCURACY RESULTS ON CIFAR-100

We run additional experiments comparing the diversity of solutions found vs their test accuracy on CIFAR-100. CIFAR-100 is an intermediate step between CIFAR-10 and ImageNet, and is overall

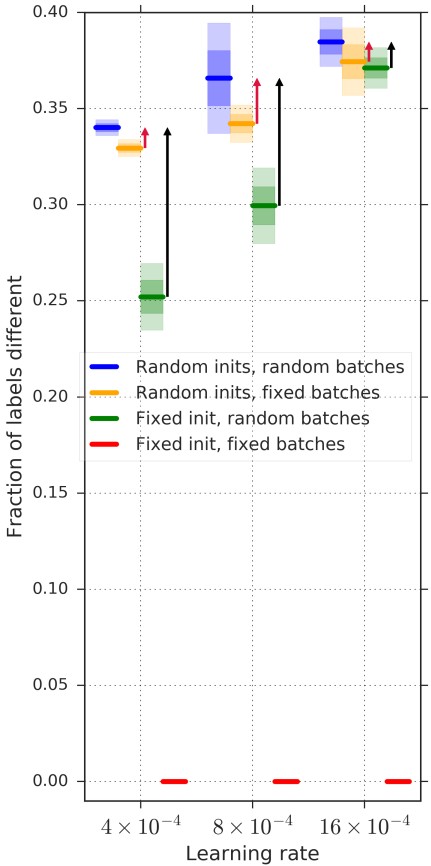

Figure S2: The effect of random initializations and random training batches on the diversity of predictions.

much more challenging to learn than CIFAR-10. Our additional results are presented in Figure S3. Solutions obtained by subspace sampling methods described in Section 4 have a worse trade off between prediction diversity (needed for ensembling) and accuracy, compared to independently initialized and trained optima. This is consistent with our results on CIFAR-10 in Figure 5.

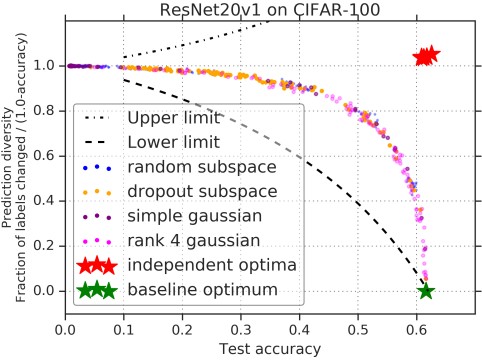

Figure S3: Diversity versus accuracy plots for a ResNet20v1 trained on CIFAR-100.

# D    DERIVING THE UPPER AND LOWER LIMIT CURVES IN THE DIVERSITY–ACCURACY PLOTS

In Figures 5 and S3 we bound our empirical results by two theoretically derived curves, limiting the expected trade off between diversity and accuracy in the best and worst case scenarios. The resulting functions are presented in the main text in Section 3.2. We will show the detailed derivations here.

Given a $C$-class classification problem and a reference solution with accuracy $a^*$, we would like to obtain a function $d_{\mathrm{diff}}(a)$ which gives us the fraction of labels on which another solution disagrees with the reference solution as a function of its accuracy.

## D.1    UNCORRELATED PREDICTIONS – THE BEST CASE

The best case scenario is when the predicted labels are uncorrelated with the reference solution's labels. On a particular example, the probability that the reference solution got it correctly is $a^*$, and the probability that our solution got it correctly is $a$. On those examples, the predictions do not differ since they both have to be equal to the ground truth label. The probability that the reference solution is correct on an example while our solution is wrong is $a^*(1 - a)$. The probability that the reference solution is wrong on an example while our solution is correct is $(1 - a^*)a$. On the examples where both solutions are wrong (probability $(1 - a^*)(1 - a)$) there are two cases: a) the two solutions agree (an additional factor of $1/(C - 1)$) or b) disagree (an additional factor of $(C - 2)/(C - 1)$). Only the case b) contributes to the fraction of labels on which they disagree. Hence we end up with

$$d_{\mathrm{diff}}(a; a^*, C) = (1 - a^*)a + (1 - a)a^* + (1 - a^*)(1 - a)\frac{C - 2}{C - 1} . \tag{3}$$

## D.2    CORRELATED PREDICTIONS – THE WORST CASE

The other extreme case is when the predictions of our new solution are just the predictions of the reference solution perturbed by perturbations of different strength. Then, the solutions retain a great amount of correlation.

Let the probability of a label changing be $p$. We will consider 4 cases: a) the label of the correctly classified image does not flip (probability $a^*(1 - p)$), b) it flips (probability $a^*p$), c) an incorrectly labelled image does not flip (probability $(1 - a^*)(1 - p)$), and d) it flips (probability $(1 - a^*)p$).

The resulting accuracy $a(p)$ obtains a contribution $a^*(1 - p)$ from case a) and with probability $1/(C - 1)$ contribution $(1 - a^*)p$ from d). Therefore $a(p) = a^*(1 - p) + p(1 - a^*)/(C - 1)$. Inverting this relationship, we get $p(a) = (C - 1)(a^* - a)/(Ca^* - 1)$. The fraction of labels on which the solutions disagree is simply $p$ by our definition of $p$, and therefore

$$d_{\mathrm{diff}}(a; a^*, C) = \frac{(C - 1)(a^* - 1a)}{Ca^* - 1} . \tag{4}$$

