# OpenReview forum: "Deep Ensembles: A Loss Landscape Perspective"
_ICLR.cc/2020/Conference — Reject_

### Official Review · AnonReviewer2 · 2019-10-23
**Official Blind Review #2**

**Rating:** 8

**Review:**

This paper is trying to answer the question why ensembles of deep neural networks trained with random initialization work so well in practice in improving accuracy. Their proposed hypothesis is that networks trained from different initializations, although all converge to a low-loss/high accuracy optimum, explore different modes in function space and therefore provide more diversity. To experimentally support their hypothesis, first they show that functions along a single training trajectory are similar, however trajectories starting from different initializations may significantly differ. The difference in function space is based on the fraction of points on which the two functions disagree in terms of their prediction. Second, they use different subspace sampling methods around a single optimum and demonstrate that they are significantly less diverse (low disagreement between predictions) than sampling from independent optima through diversity vs accuracy plots. Moreover, they comment on the recent observation that local optima are connected by low-loss tunnels. They experimentally show that even though low-loss/high accuracy path exists between local optima, these tunnels do not correspond to similar solutions in function space, further supporting the multi-mode hypothesis. The authors compare the relative benefit of subspace sampling, weight averaging and ensembling on accuracy and interpret their findings in terms of the hypothesis.

Overall, the paper is very well written and provides interesting insights into the multi-modal structure of deep neural network loss landscapes. Even though the hypothesis of the paper is not entirely new and has been touched upon in Fort & Jastrzebski (2019), this paper contributes to the field by providing thorough experimental support and clear exposition of the idea. Therefore, I would accept this paper if the authors provided additional experimental results on a different dataset.

The paper mentions that the trends are consistent across all datasets the authors have explored. However, they only provide results on CIFAR-10 (and a limited set of experiments on ImageNet). Since the contribution of the paper heavily relies on providing experimental verification, it would be important to include at least the diversity vs. accuracy plot for the other datasets they have explored to demonstrate that this phenomenon is not specific to CIFAR-10.

Additionally, I would like to add a couple of comments on the paper that are not part of my decision, but could potentially improve the paper.
-The diversity score introduced in the paper is simple and intuitive, however it would be interesting to see whether the results hold across different notions of function space disagreement.

-It is mentioned in the paper that data augmentation has been used for training the ResNet20 architecture. Would the results change significantly without data augmentation, as it adds another source of randomness to the training procedure.

-Some comments on the figures: in Figure 3 it is very difficult to discern any difference between different shades of red (disagreement values), and in this form the plots are not too informative. Maybe rescaling or a different way of presentation would help. Interpreting Figure 7/a is a bit difficult, probably a 3D plot would be useful to explain the different line sections.


**Experience Assessment:**

I have read many papers in this area.

**Review Assessment: Checking Correctness Of Derivations And Theory:**

N/A

**Review Assessment: Checking Correctness Of Experiments:**

I assessed the sensibility of the experiments.

**Review Assessment: Thoroughness In Paper Reading:**

I read the paper at least twice and used my best judgement in assessing the paper.

---

> ### Author Response · Authors · 2019-11-11
> **Response**
>
> Thank you for your detailed review and positive comments. We hope that you will champion our paper.
>
> We address some of the points you’ve made in your response.
>
> Based on your feedback, we added experiments with ResNet on CIFAR-100 which strengthened our claims and verified that other function space disagreement metrics result in the same effects.
>
> (1] We added ResNet CIFAR-100 experiments in Appendix C and they support our conclusions:
>
> We have conducted a wider range of experiments to further strengthen the validity of our claims. In particular, to make sure that the separation of the independently optimized optima and functions sampled by subspace sampling methods in the diversity-accuracy plane remain true even for more challenging datasets, we added experiments on CIFAR-100 with a ResNet. The results support our previous conclusions based on CIFAR-10 and other smaller datasets, and seem to be even stronger..
>
> (2] Different notions of function space disagreement = diversity metrics.
>
> We experimented with different distance measures between predicted probability distributions between models and settled on the fraction of predictions that are different as we thought it would be the most intuitive for the reader. We verified that the same separation between the region of independently initialized and optimized optima and the subspace sampled solutions holds for the KL-divergence, and L_n distances between the distributions (we looked at different ns, including the usual n=1 and n=2). Our conclusions, therefore, seem to be independent of the distance measure.
>
> (3] The effect of data augmentation.
>
> We have conducted experiments with both data augmentation (ResNet20v1 on CIFAR-10 and CIFAR-100) and without data augmentation (all other experiments) and the results have the same character. We added data augmentation so that our classifier accuracy is comparable with previously published results using this architecture.
>
> “Some comments on the figures:” Thank you for the suggestions, we will look into these.

---

### Official Review · AnonReviewer3 · 2019-10-23
**Official Blind Review #3**

**Rating:** 3

**Review:**

The contribution of the paper is the following two findings: 1. Despite the fact that local minima are connected in the loss landscape the functions corresponding to the points on the curve are significantly distinct. 2. The points along the training trajectory correspond to similar functions.

Originality and novelty. Both findings do not seem quite new. The first conclusion can be mostly derived from Figure 2 right [1]. Moreover, the difference between functions on the curve in terms of predictions is the main motivation of Fast Geometric Ensembling. The second conclusion is also not quite new and there were several approaches to overcome it e.g. SWA [2]. I appreciate that the authors did a much broader investigation of this phenomena than it was done in previous works. Another drawback is lack of practical implications. It is known that ensembling based on dropout is worse than independent networks, but the main advantage of this and similar approaches is memory efficiency.

The clarity. The paper is well written, contains all necessary references and is easy to follow. The provided experimental results and supporting plots are also clear and contain the necessary description. The only part that I found a bit confusing is radial plots. I would recommend the authors to add more rigorous description of how they constructed these plots to increase clarity of the paper. Can the authors please also clarify how they derived formulas for the expected fractional difference for f^* and f functions in the section 3.2?

Overall, it is an interesting paper, but the findings are not quite new.
[1]  Timur Garipov, Pavel Izmailov, Dmitrii Podoprikhin, Dmitry P Vetrov, and Andrew G Wilson. Loss surfaces, mode connectivity, and fast ensembling of DNNs. InNeurIPS, 2018
[2] Pavel Izmailov, Dmitrii Podoprikhin, Timur Garipov, Dmitry Vetrov, and Andrew Gordon Wilson. Av-eraging weights leads to wider optima and better generalization.arXiv preprint arXiv:1803.05407,2018


**Experience Assessment:**

I have published one or two papers in this area.

**Review Assessment: Checking Correctness Of Derivations And Theory:**

I carefully checked the derivations and theory.

**Review Assessment: Checking Correctness Of Experiments:**

I carefully checked the experiments.

**Review Assessment: Thoroughness In Paper Reading:**

I read the paper thoroughly.

---

> ### Author Response · Authors · 2019-11-11
> **Response**
>
> Thank you for your review.
>
> “The contribution of the paper is the following two findings: 1. Despite the fact that local minima are connected in the loss landscape the functions corresponding to the points on the curve are significantly distinct. 2. The points along the training trajectory correspond to similar functions. “
>
> These comments seem focused on particular subsections (Section 3.3 and Section 3.1) and significantly under-estimates the total contributions of our paper.
>
> To the best of our knowledge, we are the first to comprehensively investigate deep ensembles vs Bayesian neural nets from loss landscape perspective. We carefully investigated the role of random initialization in deep ensembles, tested the complementary effects of ensembling and subspace methods, and measured diversity of functions. Aside from earlier results on CIFAR-10 and ImageNet, we have also added new experiments on CIFAR-100 (see Figure S3 in Appendix C) which are consistent with our earlier results.
>
> Please see also the summary of contributions from other reviewers.
>
> R1 said “This paper analyzes ensembling methods in deep learning from the perspective of the loss landscapes. The authors empirically show that popular methods for learning Bayesian neural networks produce samples with limited diversity in the function space compared to modes of the loss found using different random initializations ... The paper also demonstrates the complementary benefits of using subspace sampling/weight averaging in combination with deep ensembles and shows that relative benefits of deep ensembles are higher. “
>
> R2 said “This paper is trying to answer the question why ensembles of deep neural networks trained with random initialization work so well in practice in improving accuracy ... Overall, the paper is very well written and provides interesting insights into the multi-modal structure of deep neural network loss landscapes.”
>
> --------------
>
> “I would recommend the authors to add more rigorous description of how they constructed these plots to increase clarity of the paper. Can the authors please also clarify how they derived formulas for the expected fractional difference for f^* and f functions in the section 3.2? “
>
> We have added the derivation of the two limiting functions in the appendix. The upper limit corresponds to the best case for ensembling, where the two functions are uncorrelated. The lower limit corresponds to the worst case, where the predictions are obtained by perturbing the outputs of the reference function by different amounts of noise, therefore retaining a large amount of correlation between their predictions. We provide the detailed derivation in the appendix our updated version.
>
> --------------
>
> “The first conclusion can be mostly derived from Figure 2 right of Garipov et al.”
>
> We do not agree that this conclusion can be reached from that figure as you are suggesting. Figure 2 in Garipov et al. only plots loss and accuracy, and does not measure function space similarity, between different initializations, or along the tunnel at all. Just by looking at accuracy and loss values, there is no way to infer how similar the predictions of the two functions are.
>
> --------------
>
> “The second conclusion is also not quite new and there were several approaches to overcome it e.g. SWA [2].”
>
> We are not sure what exactly you mean. Could you clarify your claim?
> We showed that functions along a trajectory (or subspace thereof) are similar whereas ensembling over random initializations leads to much more diversity; see sections 3.2 for diversity vs accuracy plots and Section 4 where we measure the relative effects of ensembles and subspace sampling methods. These results indicate that random initialization provides more diversity than subspace sampling methods.
>
> --------------
>
> “Another drawback is lack of practical implications. It is known that ensembling based on dropout is worse than independent networks, but the main advantage of this and similar approaches is memory efficiency.”
>
> We’re happy to add a discussion about different regimes (training time constraints, serving time constraints, memory constraints, etc), but it is beyond the scope of this paper to discuss every possible setting in detail. Some of these solutions are well-known in the literature, cf. the discussion in (Lakshminarayanan et al. 2017) or the take-home messages in (Ovadia et al. 2019): for instance, distillation is a popular solution when serving time is the primary constraint. Implicit ensembles (e.g. Monte-Carlo dropout) are popular when memory is the main constraint. The best method would obviously depend on the specific constraints (as you also point out).
>
> The goal of this work is to understand the general question of why ensembles work well and we provide an explanation from the perspective of loss landscapes. In future work, we plan to take these insights to develop better algorithms for specific settings.

---

> > ### Comment · AnonReviewer3 · 2019-11-13
> > **Response**
> >
> > I would like to thank the authors for their feedback.
> > “These comments seem focused on particular subsections (Section 3.3 and Section 3.1) and significantly under-estimates the total contributions of our paper. “
> > From my point of view, these two points that I mentioned are the concise description of the main contribution of the paper.
> > The investigation of the subspace sampling is also a significant part of the paper, but I would say that findings regarding subspace sampling somehow intersect with two items that I described.
> >
> > “The paper also demonstrates the complementary benefits of using subspace sampling/weight averaging in combination with deep ensembles and shows that relative benefits of deep ensembles are higher.“
> > I believe this conclusion can be derived based on findings from [1] (table 1), where one can interpret FGE as subspace sampling method. In this paper, FGE was combined with ensembling of models trained from different initializations. The increase of budget (e.g. using several initializations) leads to bigger improvement than a simple application of FGE. Nevertheless, combining these two approaches leads to better results, which shows that these methods can be combined.
> >
> > I would like to highlight the following one more time. The authors indeed conducted much broader investigation than was done in previous works and these results are clearly written. Nevertheless, the investigated phenomena are not quite new and this is an incremental paper. The provided conclusions are aligned with previous experiments but the authors did not provide any new insights into applications of their findings.
> > I believe if the authors add the practical application of their findings, it will significantly increase the novelty of the paper.
> >
> >
> > I would like to increase my score, but I still believe that it is a borderline paper.
> >
> > “We are not sure what exactly you mean. Could you clarify your claim? We showed that functions along a trajectory (or subspace thereof) are similar whereas ensembling over random initializations leads to much more diversity; see sections 3.2 for diversity vs accuracy plots and Section 4 where we measure the relative effects of ensembles and subspace sampling methods. These results indicate that random initialization provides more diversity than subspace sampling methods.”
> >
> > One of the main components of SWA is a cyclic learning rate schedule or constant learning rate schedule with larger learning rate than the learning rate that was used at the end of the training. If the simple averaging would be applied to the points of the training trajectory, in general it will not give a significant boost in performance. If the weights corresponding to the last epochs were taken, one would not see improvement in accuracy because the predictions have barely changed. If points were taken from the middle of the training process, one would not have seen improvement on top of the best point because the models would be much weaker. It is indeed not directly stated that predictions of the models corresponding to the same trajectory are similar, but it sounds like a fairly obvious conclusion based on it.
> >
> > [1]  Timur Garipov, Pavel Izmailov, Dmitrii Podoprikhin, Dmitry P Vetrov, and Andrew G Wilson. Loss surfaces, mode connectivity, and fast ensembling of DNNs. InNeurIPS, 2018

---

> > > ### Author Response · Authors · 2019-11-15
> > > **Response**
> > >
> > > Thank you for your response.
> > >
> > > Reading your recent comments (“The authors indeed conducted much broader investigation than was done in previous works and these results are clearly written. Nevertheless, the investigated phenomena are not quite new“) it seems like you agree that our paper conducts a much broader investigation and is a valuable contribution, but your main concerns are around novelty and discussion of practical application.
> > >
> > > We will address your specific comments below.
> > >
> > > ------
> > >
> > > Discussion of practical applications: We have added a brief discussion in our earlier comment. We will definitely add a discussion in the final version of the paper (we’re already close to the 8-page limit now).
> > >
> > > -------
> > >
> > > “From my point of view, these two points that I mentioned are the concise description of the main contribution of the paper. The investigation of the subspace sampling is also a significant part of the paper, but I would say that findings regarding subspace sampling somehow intersect with two items that I described.”
> > >
> > > We disagree with your characterization of our contribution. We believe that the main component of our paper is our comprehensive analysis of deep ensembles vs Bayesian neural nets from loss landscape perspective.
> > >
> > > As we said before, to the best of our knowledge, we are the first to comprehensively investigate deep ensembles vs Bayesian neural nets from loss landscape perspective. We carefully investigated the role of random initialization in deep ensembles, measured diversity of functions and tested the complementary effects of ensembling and subspace methods on accuracy as well as calibration under data shift.
> > >
> > > -------
> > >
> > > “I believe this conclusion can be derived based on findings from [1] (table 1), where one can interpret FGE as subspace sampling method.”
> > >
> > > Thanks for clarifying and adding a specific reference.
> > >
> > > Table 1 of [1] reports accuracy (error rate) to compare the effect of random init vs FGE.
> > >
> > > Here are some factual differences between Table 1 of [1] and our work:
> > > - It does not discuss the diversity of solutions in prediction space.
> > > - It does not present a combination of ensembles with subsampling-based Bayesian neural networks (low-rank Gaussian, diagonal Gaussian, dropout).
> > > - It also does not deal with accuracy on corrupted data, and neither does it measure  calibration under shift.
> > > - In addition to results on CIFAR-10 and CIFAR-100, we also present results on ImageNet.
> > >
> > > We like the work of [1] and we already cite [1] and related papers. We’d be happy to include a discussion about Table 1 of [1]. That said, we think it is non-trivial to derive all of our conclusions above from just the error rates reported in that table.
> > >
> > > -------
> > >
> > > “One of the main components of SWA is a cyclic learning rate schedule or constant learning rate schedule with larger learning rate than the learning rate that was used at the end of the training … It is indeed not directly stated that predictions of the models corresponding to the same trajectory are similar, but it sounds like a fairly obvious conclusion based on it.”
> > >
> > > We provide direct evidence for the similarity, see Figures 2 and 4. Whether it is "obvious" or not is a subjective question, but to us it certainly was not and that's why we investigated it.
> > >
> > > We are not the authors of the SWA paper, so we don’t really know why the SWA authors chose this particular cyclic learning rate schedule.
> > >
> > > To the best of our knowledge, the text in the SWA paper does not make a direct connection between the parameters of the cyclic learning rate, and the question of why deep ensembles work better than Bayesian neural nets.
> > >
> > > -------
> > >
> > > “Nevertheless, the investigated phenomena are not quite new and this is an incremental paper.”
> > >
> > > As we said before, we are not aware of any other work that would comprehensively study ensembles and subspace sampling methods for Bayesian neural nets and their predictions diversity from the loss landscape point of view. In particular, we discuss the specific tradeoff between accuracy and diversity of solutions that shows the clear separation between the individual optima and the subspace samples.
> > >
> > > We believe the focus of our work is sufficiently different from [1], and we believe that these papers provide complementary perspectives.
> > >
> > > It might be easy in hindsight to connect the dots between [1] and our work, especially if one adds retrospective explanations (and assumptions) not directly present in [1], e.g. R3’s comments above on:
> > > - interpreting FGE as illustrative of all subspace sampling methods for Bayesian neural networks (and assuming error on i.i.d test set reflects all performance metrics even under data shift)
> > > - connecting choice of cyclical learning rate hyperparameters used in SWA paper, to diversity vs accuracy plots in prediction space.
> > > Given the amount of additional explanations and assumptions needed beyond just the existing text in [1] to derive our conclusions, we do not think our work is “incremental”.

---

### Official Review · AnonReviewer1 · 2019-10-23
**Official Blind Review #1**

**Rating:** 3

**Review:**

This paper analyzes ensembling methods in deep learning from the perspective of the loss landscapes. The authors empirically show that popular methods for learning Bayesian neural networks produce samples with limited diversity in the function space compared to modes of the loss found using different random initializations. The paper also considers the low-loss paths connecting independent local optima in the weight-space. The analysis shows that while the values of the loss and accuracy are nearly constant along the paths, the models corresponding to different points on a path define different functions with diverse predictions. The paper also demonstrates the complementary benefits of using subspace sampling/weight averaging in combination with deep ensembles and shows that relative benefits of deep ensembles are higher.

The paper is well-written. The experiments are described well and the results are presented clearly in highly-detailed and visually-appealing figures. There are occasional statements which are not formulated rigorously enough (see comments below).

The paper presents a thorough experimental study of different ensemble types, their performance, and function space diversity of individual members of an ensemble. In my view, the strongest contribution of the paper is the analysis of the diversity of the predictions for different sampling procedures in comparison to deep ensembles. However, the novelty and the significance of the other contributions are limited (see comments below). Therefore, I consider the paper to be below the acceptance threshold.

Comments and questions to authors:
1) The practical aspects of different ensembling techniques are not discussed in the paper. While it is known that deep ensembles generally demonstrate stronger performance [1], there is a trade-off between the ensemble performance and training time/memory consumption. The considered alternative ensembling procedures can be favorable in specialized settings (e.g. limited training time and/or memory).

2) It remains unclear to me what new insights does the analysis of the low-loss connectors provide? It is expected (and in fact can be shown analytically) that if the two modes define different functions then intermediate points on a continuous path define functions which are different from those defined by the end-points of the path. This result was also analyzed before from the perspective of the performance of ensembles formed by the intermediate points on the connecting paths (see Fig. 2 right in [2]).
Moreover, I would encourage authors to reformulate the statements on the connectivity in the function space such as:
-- “We demonstrate that while low-loss connectors between modes exist, they are not connected in the space of predictions.”  (Abstract)
-- “the connectivity in the loss landscape does not imply connectivity in the space of functions” (Discussion)
In my opinion, these claims are somewhat misleading. What does it mean that the modes are disconnected in the function space? Neural networks define continuous functions (w.r.t to both the inputs and the weights), and a connector is continuous path in the weight space which continuously connects the modes in the function space (i.e. a path defines a homotopy between two functions). It is true that two modes correspond to two different functions. However, it is unclear in which sense these functions can be considered to be disconnected.

[1] Balaji Lakshminarayanan, Alexander Pritzel, and Charles Blundell. Simple and scalable predictive uncertainty estimation using deep ensembles. In NeurIPS, 2017.

[2] Timur Garipov, Pavel Izmailov, Dmitrii Podoprikhin, Dmitry P Vetrov, and Andrew G Wilson. Loss surfaces, mode connectivity, and fast ensembling of DNNs. In NeurIPS, 2018.

**Experience Assessment:**

I have published one or two papers in this area.

**Review Assessment: Checking Correctness Of Derivations And Theory:**

N/A

**Review Assessment: Checking Correctness Of Experiments:**

I carefully checked the experiments.

**Review Assessment: Thoroughness In Paper Reading:**

I read the paper thoroughly.

---

> ### Author Response · Authors · 2019-11-11
> **Response**
>
> Thank you for your review and the positive comments about our work.
> We would like to address the points you brought up.
>
> “Practical aspects of ensembling versus different methods of subspace sampling”:
>
> The goal of this work is to understand the general question of why ensembles work well and we provide an explanation from the perspective of loss landscapes. In future work, we plan to take these insights to develop better algorithms for specific settings.
> We’re happy to add a discussion about different regimes (training time constraints, serving time constraints, memory constraints, etc), but it is beyond the scope of this paper to discuss every possible setting in detail. Some of these solutions are well-known in the literature, cf. the discussion in (Lakshminarayanan et al. 2017) or the take-home messages in (Ovadia et al. 2019): for instance, distillation is a popular solution when serving time is the primary constraint. Implicit ensembles (e.g. Monte-Carlo dropout) are popular when memory is the main constraint. The best method would obviously depend on the specific constraints (as you also point out).
>
> ----------------
>
> “It remains unclear to me what new insights does the analysis of the low-loss connectors provide?”
>
> We added Section 3.3 in response to feedback on an earlier version of this paper. A couple of folks thought that our results contradicted the results from earlier papers on “mode connectivity”. We believe this confusion comes down to how folks interpret the word “connectivity”. The original papers by (Garipov et al. 2018) and (Draxler et al. 2018) used “connectivity” to imply continuous map between two functions (the notion you mentioned), but others (not the original authors) seem to have interpreted connectivity as similarity of functions.
> We mainly wanted to convey that identical loss values do not imply identical functions. That is, loss similarity, which measures if L(f_{theta_1}) and L(f_{theta_2}) are similar, does not measure prediction similarity, which measures if f_{theta_1} and f_{theta_2} are similar.
> While it has been shown that two independently initialized and optimized-to optima can in fact be connected on a low-loss path in the weight space by (Garipov et al. 2018) and (Draxler et al. 2018), the papers do not explicitly discuss how similar the models along such a path are in their predictions, which can be taken as a proxy for their similarity in the space of functions.
> Given that multiple folks raised this point about “connectivity”, we thought it might be useful to explicitly add a discussion about the distinction between loss similarity and prediction similarity in subsection 3.3.
>
> ----------------
>
> “I would encourage authors to reformulate the statements on the connectivity in the function space”:
>
> We can rephrase “loss connectivity” and “function space connectivity” to “loss similarity” and “predictions similarity”, would that address your concerns?
>
> ----------------
>
> “However, the novelty and the significance of the other contributions are limited (see comments below). Therefore, I consider the paper to be below the acceptance threshold.”
>
> To the best of our knowledge, we are the first to comprehensively investigate deep ensembles vs Bayesian neural nets from loss landscape perspective. We carefully investigated the role of random initialization in deep ensembles, tested the complementary effects of ensembling and subspace methods, and measured diversity of functions. Aside from earlier results on CIFAR-10 and ImageNet, we have also added new experiments on CIFAR-100 (see Figure S3 in Appendix C) which are consistent with our earlier results.
>
> I think your own summary highlights a lot of our contributions: “This paper analyzes ensembling methods in deep learning from the perspective of the loss landscapes. The authors empirically show that popular methods for learning Bayesian neural networks produce samples with limited diversity in the function space compared to modes of the loss found using different random initializations ... The paper also demonstrates the complementary benefits of using subspace sampling/weight averaging in combination with deep ensembles and shows that relative benefits of deep ensembles are higher. “
>
> We believe these results are both novel and significant, and would be interesting to the ICLR community.

---

### Decision · Program_Chairs · 2019-12-19

**Decision:**

Reject

**Comment:**

Paper https://arxiv.org/abs/1802.10026 (Garipov et. al, NeurIPS 2018) shows that one can find curves between two independently trained solutions along which the loss is relatively constant. The authors of this ICLR submission claim as a key contribution that they show the weights along the path correspond to different models that make different predictions ("Note that prior work on loss landscapes has focused on mode-connectivity and low-loss tunnels, but has not explicitly focused on how diverse the functions from different modes are, beyond an initial exploration in Fort & Jastrzebski (2019)"). Much of the disagreement between two of the reviewers and the authors is whether this point had already been shown in 1802.10026.

It is in fact very clear that 1802.10026 shows that different points on the curve correspond to diverse functions. Figure 2 (right) of this paper shows the test error of an _ensemble_ of predictions made by the network for the parameters at one end of the curve, and the network described by \phi_\theta(t) at some point t along the curve: since the error goes down and changes significantly as t varies, the functions corresponding to different parameter settings along these curves must be diverse. This functional diversity is also made explicit multiple times in 1802.10026, which clearly says that this result shows that the curves contain meaningfully different representations.

In response to R3, the authors incorrectly claim that  "Figure 2 in Garipov et al. only plots loss and accuracy, and does not measure function space similarity, between different initializations, or along the tunnel at all. Just by looking at accuracy and loss values, there is no way to infer how similar the predictions of the two functions are." But Figure 2 (right) is actually showing the test error of an average of predictions of networks with parameters at different points along the curve, how it changes as one moves along the curve, and the improved accuracy of the ensemble over using one of the endpoints. If the functions associated with different parameters along the curve were the same, averaging their predictions would not help performance.

Moreover, Figure 6 (bottom left, dashed lines) in the appendix of 1802.10026 shows the improvement in performance in ensembling points along the curve over ensembling independently trained networks. Section A6 (Appendix) also describes ensembling along the curve in some detail, with several quantitative results. There is no sense in ensembling models along the curve if they were the same model.

These results unequivocally demonstrate that the points on the curve have functional diversity, and this connection is made explicit multiple times in 1802.10026 with the claim of meaningfully different representations: “This result also demonstrates that these curves do not exist only due to degenerate parametrizations of the network (such as rescaling on either side of a ReLU); instead, points along the curve correspond to meaningfully different representations of the data that can be ensembled for improved performance.”  Additionally, other published work has built on this observation, such as 1907.07504 (UAI 2019), which performs Bayesian model averaging over the mode connecting subspace, relying on diversity of functions in this space; that work also visualizes the different functions arising in this space.

It is incorrect to attribute these findings to Fort & Jastrzebski (2019) or the current submission.  It is a positive contribution to build on prior work, but what is prior work and what is new should be accurately characterized, and currently is not, even after the discussion phase where multiple reviewers raised the same concern. Reviewers appreciated the broader investigation of diversity and its effect on ensembling, and the more detailed study regarding connecting curves. In addition to the concerns about inaccurate claims regarding prior work and novelty (which included aspects of the mode connectivity work but also other works), several reviewers also felt that the time-accuracy trade-offs of deep ensembles relative to standard approaches were not clearly presented, and comparisons were lacking. It would be simple and informative to do an experiment showing a runtime-accuracy trade-off curve for deep ensembles alongside FGE and various Bayesian deep learning methods and mc-dropout. It's also possible to use for example parallel MCMC chains to explore multiple quite different modes like deep ensembles but for Bayesian deep learning. For the paper to be accepted, it would need significant revisions, correcting the accuracy of claims, and providing such experiments.